# Diagnostic Value of Neutrophil CD64 in Sepsis Patients in the Intensive Care Unit: A Cross-Sectional Study

**DOI:** 10.3390/diagnostics13081427

**Published:** 2023-04-15

**Authors:** Huy Minh Pham, Duy Ly Minh Nguyen, Minh Cuong Duong, Linh Thanh Tran, Thao Thi Ngoc Pham

**Affiliations:** 1Department of Emergency and Critical Care, Faculty of Medicine, University of Medicine and Pharmacy at Ho Chi Minh City, Ho Chi Minh City 72714, Vietnam; 2Intensive Care Unit, Cho Ray Hospital, 201B Nguyen Chi Thanh Street, District 5, Ho Chi Minh City 72713, Vietnam; 3School of Population Health, University of New South Wales, Sydney, NSW 2052, Australia

**Keywords:** nCD64, sepsis, early diagnosis, intensive care unit, ICU

## Abstract

Little is known about the role of neutrophil CD64 (nCD64) in detecting sepsis early in Asian populations. We examined the cut-off and predictive values of nCD64 for diagnosing sepsis in Vietnamese intensive care units (ICU) patients. A cross-sectional study was conducted at the ICU of Cho Ray Hospital between January 2019 and April 2020. All 104 newly admitted patients were included. Sensitivity (Sens), specificity (Spec), positive and negative predictive values (PPV and NPV), and receiver operating characteristic (ROC) curves were calculated to compare the diagnostic values of nCD64 with those of procalcitonin (PCT) and white blood cell (WBC) for sepsis. The median nCD64 value in sepsis patients was statistically higher than that of non-sepsis patients (3106 [1970–5200] vs. 745 [458–906] molecules/cell, *p* < 0.001). ROC analysis found that the AUC value of nCD64 was 0.92, which was higher than that of PCT (0.872), WBC (0.637), and nCD64 combined, with WBC (0.906) and nCD64 combined with WBC and PCT (0.919), but lower than that of nCD64 combined with PCT (0.924). With an AUC value of 0.92, the nCD64 index of 1311 molecules/cell-detected sepsis with 89.9% Sens, 85.7% Spec, 92.5% PPV, and 81.1% NPV. nCD64 can be a useful marker for early sepsis diagnosis in ICU patients. nCD64 combined with PCT may improve the diagnostic accuracy.

## 1. Introduction

Sepsis remains the most common cause of death among patients in intensive care units (ICUs) worldwide [1]. Sepsis is defined as a dysregulated host response to infection, which results in life-threatening organ failure [2]. Despite several improvements in understanding the pathogenesis of sepsis, the associated mortality rate is still high. A large systematic review and meta-analysis of 170 studies conducted in Europe, North America, and Australia between 2009 and 2019 found that the 90-day mortality of septic shock and sepsis was 38.47% and 32.24%, respectively [3]. The prospective MOSAICS II study conducted at 386 ICUs across 22 Asian countries (including Vietnam) also showed that the overall prevalence of sepsis in ICUs was 22.4%, and the hospital mortality rate of sepsis was 32.6% [4]. It has been well documented that a delay in diagnosis and management of sepsis will cause an increase in mortality and morbidity [5]. Therefore, early detection and adequate treatment in the first few hours after the development of sepsis can improve outcomes [6].

Distinguishing between a severe sepsis that can lead to septic shock and a non-infectious systemic inflammatory response syndrome is difficult. It is because the common clinical signs of severe sepsis including fever, tachycardia, and tachypnea can be detected in several non-infectious diseases [7]. In addition, the laboratory tests are not usually specific for sepsis, while microbiological evidence of sepsis is often obtained at a late stage [8]. Regarding laboratory tests, several biomarkers such as TNFα, IL-1, IL-10, IL-6, serum procalcitonin (PCT), and C-reactive protein (CRP) are currently being used to detect sepsis [9]. However, these cytokines have short half-lives, poor stability, and strict storage requirements [10]. In particular, PCT and CRP have poor specific diagnostic value and are not reliable [11]. The white blood cell (WBC) count which are also widely used to screen sepsis are poor predictors of sepsis [12]. Recently, a new group of sepsis biomarkers has also been studied and includes microRNAs (miRNA or miR), which are produced from cells and subsequently released into the circulation during inflammation, infection, and sepsis [13]. A recent meta-analysis showed that miRNAs, particularly miRNA-155-5p, with a pooled sensitivity (Sens) of 71%, a pooled specificity (Spec) of 82%, and area under the ROC curve (AUC) of 0.85, may serve as helpful indicators of sepsis [14]. However, some major limitations associated with miRNAs prevent them from being widely used at this stage, especially in low-resource settings. There is a controversy over the use of serum or plasma samples, all of which have been shown to be a source of heterogeneity of miRNA expression [14]. A standardized measurement method has also not been established [15]. In addition, most miRNA detection methods are time consuming and costly [15]. Thus, there is a critical need for a biomarker that can be widely used to detect sepsis at an early stage so that an appropriate antibiotic therapy can be started in a timely manner [9].

It has been found that dysregulation of innate and adaptive immunity is a significant contributor in the pathogenesis and development of sepsis. Neutrophil CD64 (nCD64) is a high-affinity immunoglobulin Fc-γ receptor I [16]. The nCD64 expression has been studied for years as a biomarker of infection and sepsis because of its reported low baseline expression and quick increase after inflammation [16]. Regarding the diagnostic value of the nCD64 index for diagnosing infection in adult patients, a systematic review and meta-analysis of eight studies comprising 1986 patients by Wang et al. found that the pooled sensitivity and specificity were 76% (95%CI: 73–78%) and 85% (95%CI: 82–87%), respectively, indicating that measuring nCD64 expression values is beneficial for early diagnosing sepsis in critically ill patients [17]. In another more recent systematic review and meta-analysis of 14 studies comprising 2471 patients in 2019, Chun-Fu Yeh et al. [18] compared the accurate diagnosis levels of nCD64, PCT, and CRP for sepsis. The study showed that in adult patients with sepsis, the nCD64 index is an excellent biomarker with a diagnostic accuracy outperforming both CRP and PCT determinations [18]. It should be noted that in these reviews, all included studies were conducted in western countries, except two studies from China. This reflects the scarcity of knowledge about the diagnostic values of nCD64 in patients with sepsis in Asia including Vietnam. This study aimed to examine the role of the nCD64 index in diagnosing sepsis in adult ICU patients by comparing the accurate diagnosis levels of nCD64 with PCT and WBC. The study also evaluated changes in the accurate diagnostic levels of nCD64 when it is used in combination with PCT and WBC.

## 2. Materials and Methods

### 2.1. Study Context and Design

A cross-sectional study was conducted at the General ICU of Cho Ray Hospital (CRH), Ho Chi Minh City, Vietnam from January 2019 to April 2020. CRH with 2300 beds is the largest, tertiary, general hospital and is the last line of the medical treatment system in southern Vietnam [19]. CRH has four ICUs including the General ICU where the study was conducted. The General ICU has 28 beds and receives three to four patients per day. Most patients are those with multiple traumas, sepsis, or septic shock. During the study period, all adult patients admitted to the General ICU were screened based on the inclusion and exclusion criteria and invited to participate in the study. Informed consent was obtained from patients or their direct caregivers before patients participated in the study. The study was conducted in accordance with the Declaration of Helsinki, and the protocol was approved by the Ethics Committee of the University of Medicine and Pharmacy at Ho Chi Minh City (approval number 103/ĐHYD-HĐĐĐ).

The inclusion criteria included those with sepsis or without sepsis. Sepsis patients were defined as patients who had a clear source of infection and met the diagnostic criteria for sepsis in accordance with the third International Consensus Definitions for Sepsis and Septic Shock (Sepsis-3) [2]. Non-sepsis patients were defined as those who were in need for ICU admissions during the same period, but without immediate concern about sepsis, such as those patients admitted to the General ICU due to non-infectious health conditions such as cerebrovascular accident, acute myocarditis, poisoning, or electric shock. The exclusion criteria included patients <16 years old, refusal to give informed consent to participate in the study, and patients with a malignant disease.

The total sample size was calculated using the formula based on sensitivity [20]. Previous studies by Sakr et al. [21] and Dimoula et al. [22] showed that the prevalence of sepsis in other comparable ICUs in Southeast Asia was 39.3%, and the estimated sensitivity of the nCD64 index was 89%, respectively. Hence, with an absolute error of 10% and a type I error of 0.05, the minimum total sample size was 97 (with minimum number of sepsis cases at 38).

A questionnaire was used to collect the study participants’ information on ICU admission, including demographic characteristics (age, sex, and body mass index (BMI)), comorbidities, clinical signs (blood pressure, respiratory rate, and pulse rate), severity scores (Sequential Organ Failure Assessment (SOFA) and Acute Physiology and Chronic Health Evaluation II (APACHE II) scores [23,24]), laboratory tests, and treatment outcomes. SOFA is a scoring system that examines the sequence of complications of critical illness related to organ systems including liver, lungs, blood, hemodynamic, kidney, and neurologic. Each organ system is assigned a score ranging from 0 (normal) to 4 (high degree of dysfunction/failure) [24]. APACHE II is a measure of disease severity based on 12 current physiologic variables, previous health conditions, and age. The total score ranges from 0 to 71, which is corresponding to the increasing risk of hospital death [23]. Laboratory tests included complete blood count including WBC count, coagulation test, serum lactate, PCT, and the nCD64 index. Treatment outcomes included mortality, ventilator days, and ICU and hospital length of stay.

### 2.2. Measurement of nCD64 Levels

The nCD64 index was measured by the BD FACS CANTO system (Becton Dickinson, San Jose, CA, USA) which uses a phycoerythrin (PE) fluorescence quantification kit (Quanti BRITE PE, Becton Dickinson) [25]. Blood samples were drawn for flow cytometry analysis and kept in EDTA vials at 2–8 °C before being processed within four hours. Fifty μL of whole blood were incubated with anti-CD14-FITC (clone MφP9), anti-CD64-PE (clone MD22), and CD45PerCP (clone 2D1) for 30 min at room temperature. After the red blood cell lysis, the blood samples were washed, and the remaining cells were resuspended in sheath fluid. A calibration curve was generated using QuantiBRITE PE beads. Each QuantiBRITE PE tube contains lyophilized pellet of beads conjugated with four known levels of phycoerythrin molecules, enabling the construction of a standard curve for calculating the mean number of PE molecules on a cell. Based on the Quanti BRITE PE calibration beads with known numbers of PE molecules, the inter-assay standardization for nCD64 quantitation was carried out. The median fluorescence intensity of the respective Quantibrite PE beads assisted in the calculation of the antibody bound per cell values for nCD64. Data analysis was performed using the BD FACS Diva software version 6.1.3. This process has been validated elsewhere [26,27,28]. All laboratory tests were performed at the standardized Laboratory Department of CRH.

### 2.3. Statistical Analysis

All statistical analysis was performed using the R Statistical Software (version 3.6.2). Categorical variables were presented as an absolute count and percentage and were analyzed using Pearson’s chi-square test. Continuous variables were first tested for normality using Shapiro–Wilk test. Variables with a normal distribution are reported as mean ± standard deviation (SD) and median (25th–75th percentile) otherwise. Between-group comparisons of continuous variables were conducted using Student’s *t*-test for independent data if normally distributed and Mann–Whitney U test otherwise. Receiver operating characteristic (ROC) analysis was conducted to compare the performance of the nCD64 index with that of other measures (PCT and WBC) in the identification of sepsis. Sens, spec, positive predictive value (PPV), and negative predictive value (NPV) of the cutoff values of these biomarkers in detecting sepsis were calculated. The cut-off values were determined based on the Youden’s index, which maximizes the sum of the sensitivity and specificity (J = max [sensitivity + specificity-1]). Statistical significance was set at *p* < 0.05.

## 3. Results

### 3.1. Baseline Characteristics of Study Participants

During the study period, there was a total of 108 eligible study participants, of whom four were excluded (Figure 1). Hence, the final number of study participants was 104, including 69 patients with sepsis and 35 without sepsis.

Among 104 study participants, the mean BMI was 23.2 ± 3.21 kg/m^2^ and the medium serum lactate was 2.20 [1.48; 4.12] mmol/L (Table 1). Compared with the non-sepsis group, the sepsis group was statistically older (61.3 ± 16.3 vs. 40.6 ± 15.8, *p* < 0.001) and had significantly a lower proportion of males (47.8% vs. 77.1%, *p* = 0.008), higher median APACHE II scores (21.0 [17.0; 25.0] vs. 15.0 [8.0; 18.0]; *p* < 0.001) and mean SOFA scores (9.22 ± 3.97 vs. 4.97 ± 2.65; *p* < 0.001). Thirty patients (43.5%) in the sepsis group died, compared with seven patients (20%) in the non-sepsis group (*p* = 0.02).

The median nCD64 levels of the sepsis group of 3106 [1970; 5200] molecules/cell were statistically higher than that (745 [458; 906] molecules/cell) of the non-sepsis group (*p* < 0.001) (Table 2 and Figure 2). However, while the median PCT levels were similarly statistically higher in the sepsis group (1.10 [0.30; 4.63] ng/mL vs. 43.4 [4.97; 70.30] ng/mL; *p* < 0.001), the median WBC counts were statistically higher in the non-sepsis group (17.3 [13.1; 20.4] G/L vs. 13.9 [10.2; 18.5] G/L; *p* = 0.023).

### 3.2. The Predictive Values of nCD64, PCT, and WBC and Combinations of These Parameters

The area under the ROC curve (AUC) value of using nCD64 alone was 0.920, which was higher than that of PCT (0.872), WBC (0.637), and nCD64 combined with WBC (0.906), and a combination of nCD64, WBC, and PCT (0.919) but lower than that of nCD64 combined with PCT (0.924) (Table 3 and Figure 3). From the ROC curve of using nCD64 alone, the optimal cutoff point of the nCD64 index was calculated as 1311 molecules/cell to predict sepsis with a Sens of 89.9%, Spec of 85.7%, PPV of 92.5%, and NPV of 81.1%. Similarly, from the ROC curve of using PCT alone, the optimal cutoff point of PCT was calculated as 13.385 ng/mL to predict sepsis with a Sens of 65.2%, Spec of 93.9%, PPV of 95.7%, NPV of 56.4%. nCD64 combined with PCT could predict sepsis with a Sens of 88.4%, Spec of 87.9%, PPV of 93.8%, and NPV of 78.4%. The AUC of using WBC alone was presented in Table 3.

## 4. Discussion

In our study, patients with sepsis were more severe than those without sepsis demonstrated by higher APACHE II and SOFA scores being documented in the sepsis group. The reason is that our hospital is the last line of the medical treatment system in southern Vietnam [19]. Hence, septic patients receiving treatment at our study clinic are usually severe patients with multi-organ failure and, thus, come with higher APHACHE II and SOFA scores. In contrast, based on our experience, our non-septic patients are those with a cerebrovascular accident, acute myocarditis, poisoning, or electric shock. Although these patients need to be admitted to the ICU, they do not develop multi-organ failure, and thus have low APACHE II and SOFA scores. Regarding the severity of septic patients, in a previous Vietnamese study conducted at the same study clinic, sepsis patients also had comparable mean APACHE II and SOFA scores of 23.3 ± 8.3 and 10.6 ± 3.6, respectively [29]. Similarly, in a Chinese study by Ye et al., the mean APACHE II and SOFA scores of patients with sepsis (21.17 ± 8.34 and 11.26 ± 5.95) were statistically higher than those of patients without sepsis (12.00 ± 3.2 and 6.56 ± 2.74) [30]. As discussed previously, this observation can be attributable to the standard definition of sepsis which is an organ dysfunction that can be identified as an acute change in total SOFA score ≥2 points consequent to the infection [2]. In parallel with the SOFA score, the APACHE II score which consists of three parts including Acute Physiology Score, age points, and chronic health points are also higher in sepsis patients [23]. Consequently, due to such high severity scores, our patients with sepsis had a statistically significantly higher in-hospital mortality rate compared with those without sepsis (43.5% vs. 20%).

In the ICU settings, a number of biomarkers have been well studied as diagnostic tools for early detection of sepsis such as PCT, CRP, and WBC [9]. An ideal biomarker for sepsis in these settings would enhance the initial diagnostic evaluation of patients presenting with an infection and thus, facilitate prompt initiation of antibiotic therapy [31]. More recently, there has been an increased interest in the quantification of nCD64 expression as an early indicator of sepsis in adult patients [30,32,33]. The low expression in healthy individuals, rapid elevation in response to inflammatory cytokine stimulation, and biochemical stability at room temperature for more than 30 h of nCD64 represent its advantages as a diagnostic marker for sepsis in ICU settings [34]. In our study, 69 sepsis patients had a median nCD64 index of 3106 (1970–5200) molecules/cell. Our result is consistent with that of previous studies conducted on sepsis patients using the same Flow Cytometry technique, such as the study by Cardelli et al. in 2008 with a median value of nCD64 of 4226 (2887–6845) molecules/cell [35] and the study by Righi et al. in 2014 with the median value of nCD64 of 3842 (2799–5283) molecules/cell [36]. In addition, similar to our findings, these studies also found that the levels of nCD64 expression in patients with sepsis were higher than those observed in patients without sepsis [35,36]. In addition to nCD64, we found that both PCT and WBC have significant differences between the two groups of sepsis status. While the sepsis patients’ median PCT value of 43.4 ng/mL (4.97–70.30) was statistically significantly higher than that of the non-sepsis patients (1.10 ng/mL, [0.30–4.63]), the WBC levels in the sepsis group (13.9 G/L, [10.2–18.5]) were lower than those of the non-sepsis patients (17.3 G/L, [13.1–20.4]). However, we found a higher AUC value of nCD64 (0.92) compared with that of PCT (0.872) and WBC (0.637). This demonstrates the superior performance of nCD64 in detecting sepsis in our study population. In a recent systematic review and meta-analysis conducted by Yeh et al., the nCD64 index with an AUC of 0.94 was proven as an excellent biomarker with moderate accuracy, outperforming PCT determinations that had an AUC of 0.89 [18]. Similarly, Cui et al. reported that among four markers, including nCD64, PCT, CRP and WBC, the nCD64 index had the highest AUC value (0.91 vs. 0.79, 0.68 and 0.6, respectively), demonstrating the best diagnostic value of nCD64 for sepsis [37]. In light of these studies and our findings, we believe that nCD64 is a useful sepsis screening tool for ICU patients.

Our study found that the sensitivity of the nCD64 index was higher than that of PCT and WBC. At a cut-off point of 1311 molecules/cell, the nCD64 expression had a sensitivity of 89.9% and a specificity of 85.7% as compared with those of PCT (sensitivity: 65.2%; specificity: 93.9%) and WBC (sensitivity: 73.9%; specificity: 54.3%) for diagnosing sepsis at ICU admission. This suggests that the nCD64 index is a good diagnostic marker. These results align with previous studies in critically ill adult patients in ICU settings [22,38,39]. In detail, Bauer et al. reported that nCD64 can correctly diagnose confirmed infection at a cut-off of 1084 molecules/cell with AUC, sensitivity, and specificity of 0.86, 79.8% and 80.2%, respectively [38]. In another study by Dimoula et al., the nCD64 index had a sensitivity of 89% and a specificity of 87% at a cut-off of 230 Median fluorescence intensity (MFI)and AUC of 0.94 [22]. Dimoula et al. also recommended that daily monitoring of the nCD64 expression in critically ill patients may help diagnose ICU-acquired infection, thereby facilitating a timely antibiotic therapy [22]. In clinical practice, a biomarker with high sensitivity could be served as a tool to “rule out” a disease of interest [40]. Considering this, the nCD64 index, which is more sensitive than other markers, could be a tool to detect sepsis in patients with a low-to-intermediate probability of having sepsis.

In another most recent systematic review and meta-analysis of 54 studies comprising 9842 patients in 2021, Cong et al. found that despite the high performance of nCD64, the cut-off values of this marker varied between the included studies [39]. The differences in laboratory methodology, time of sampling, and age of participants may explain the discrepancies in the cut-off values and AUCs reported in all these studies [39]. Therefore, it is strongly suggested that future studies are needed to determine the optimal cut-off value of this biomarker in different populations [39]. The findings of our study contribute to the determination of the cut-off of nCD64 in diagnosing sepsis in ICU patients in Vietnam. Despite this, we believe that more large studies are needed to understand the variance of the cut-off values of nCD64 in diagnosing sepsis in the wider Vietnamese population.

Given that sepsis is a complex, dynamic syndrome, no single test is sufficiently sensitive and specific for detecting sepsis [17]. Hence, several studies have suggested that the use of combinations of different biomarkers is a practical approach to improve the accuracy of diagnosing sepsis [41]. Gilbot et al. found that compared with the sole use of nCD64, PCT, and sTREM-1, the “bioscore” that combined all these three markers had the highest performance in diagnosing sepsis in critically ill patients [42]. Similarly, Bauer et al. proved that a combination of nCD64, CRP, and PCT was a significant predictor of sepsis with a high AUC of 0.90 that increased sepsis diagnosis accuracy [38]. Jamsa et al. examined three biomarkers including CRP, PCT, and nCD64 individually and in combination with each other regarding their ability to identify sepsis in adult ICU patients [33]. The nCD64 index was found to be superior to the other two biomarkers alone, and there was an improved diagnostic accuracy when nCD64 was analyzed simultaneously with positive CRP and PCT [33]. In a meta-analysis of eight studies comprising 1986 patients in 2015, Wang et al. also recommended combining nCD64 with other sepsis biomarkers to help improve the accuracy of diagnosis [17]. In line with the findings of these studies, our study found that the AUC (0.924) of nCD64 combined with PCT was higher than that of any other parameter alone or in combination with each other. Therefore, in clinical practice, given the high performance of nCD64 in diagnosing sepsis, the addition of PCT will further improve the accuracy of diagnosis.

## 5. Limitations

Our study has some limitations. Given the single-center study design, the study findings may not be generalizable. In addition to the exclusion criteria, we did not recruit suspected sepsis patients whose diagnosis was not confirmed, those with end-stage chronic diseases, and pediatric patients receiving treatment at our study clinic. The number of eligible patients also sharply dropped during the last stage of the study period from January 2020 to April 2020, because the study clinic became a COVID-19 designated treatment center at this time. Hence, although a sufficient sample size was secured, the diversity of our study participants may be limited. CRP was not performed on our study participants due to financial constraints and local policies. Although CRP is commonly used for early diagnosis of sepsis in several studies [43,44], we were unable to concurrently measure CRP and PCT according to the health insurance-related restrictions in Vietnam. Other sepsis markers including miRNAs, sTREM-1, and Pentraxin-3 (PTX3) were also not examined in our study due to financial constraints [9,15]. There were differences in the severity of health condition between those with and without sepsis in our study, which can also be found in other similar studies worldwide [30,32,38,45]. This may induce bias in assessing the effectiveness of nCD64 in diagnosing sepsis. Nevertheless, to the best of our knowledge, our study is the first study in Vietnam to assess the efficacy of the nCD64 index in diagnosing sepsis in ICU patients. Our study is also among the few studies using the latest definition of sepsis (sepsis-3) in examining the diagnostic value of nCD64 [2].

## 6. Conclusions

The nCD64 index is a useful screening tool for the early detection of sepsis in adult patients admitted to the ICU, especially those with low-to-intermediate risk of acquiring sepsis. A combination of nCD64 and PCT would further enhance diagnostic accuracy. More large studies are crucial to understand the variance of the cut-off values of the nCD64 index for diagnosing sepsis in the wider Vietnamese population and comparable countries.

## Figures and Tables

**Figure 1 diagnostics-13-01427-f001:**
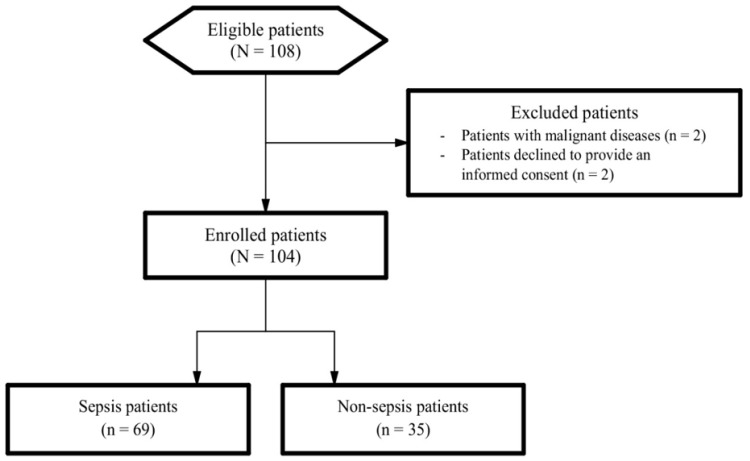
Flow chart of eligible and enrolled patients.

**Figure 2 diagnostics-13-01427-f002:**
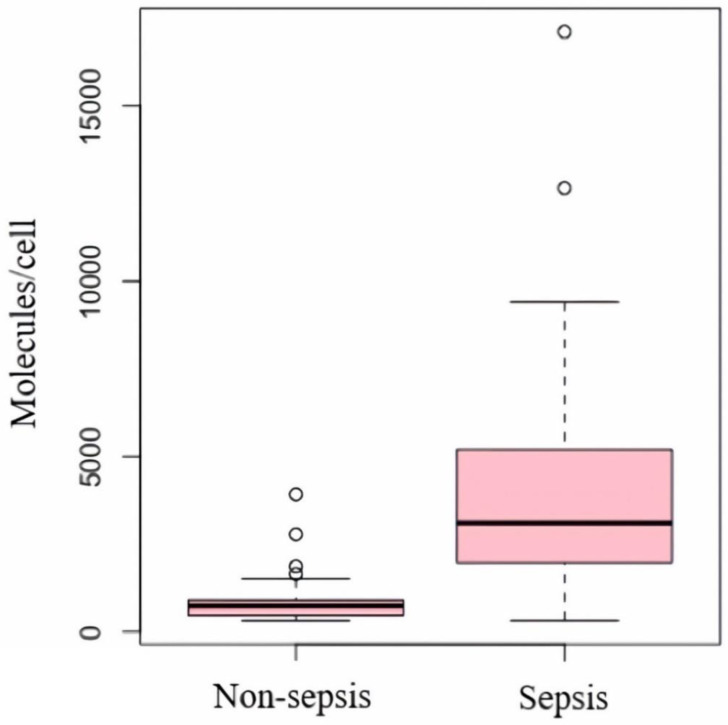
Box plot of the distribution of nCD64 values in the sepsis and non-sepsis groups.

**Figure 3 diagnostics-13-01427-f003:**
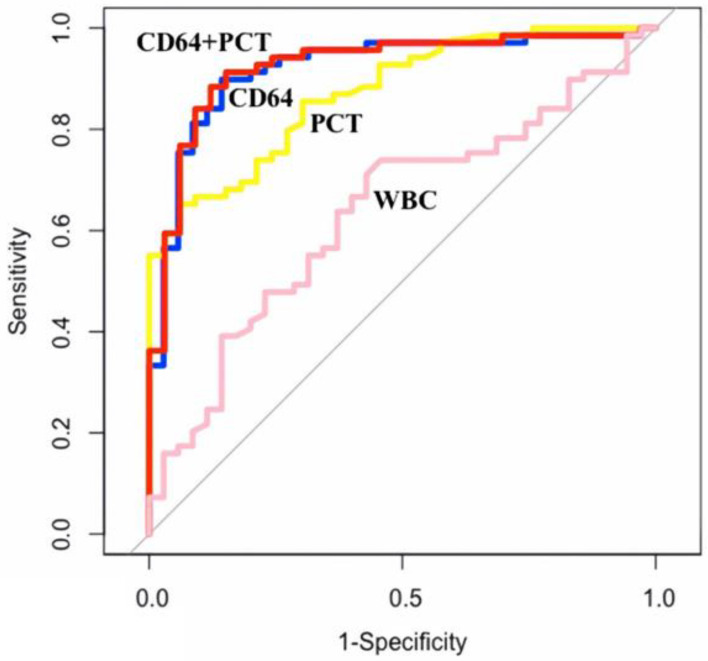
ROC curves comparing sepsis diagnostic abilities of nCD64 to PCT, WBC, and nCD64 combined with PCT.

**Table 1 diagnostics-13-01427-t001:** General characteristics of 104 study participants.

Characteristics	Non-Sepsis Group	Sepsis Group	Total Population	*p*
	*n* = 35	*n* = 69	*n* = 104	OR (95%CI)
Age (years)	40.6 ± 15.8	61.3 ± 16.3	54.3 ± 18.9	<0.001 ^t^
Male	27 (77.1)	33 (47.8)	60 (57.7)	0.008 ^c^ 0.27 (0.11–0.68)
BMI (kg/m^2^)	23.0 ± 2.8	23.2 ± 3.42	23.2 ± 3.21	0.778 ^t^
APACHE II (score)	15.0 [8.0; 18.0]	21.0 [17.0; 25.0]	19.0 [15.0; 23.0]	<0.001 ^u^
SOFA (score)	4.97 ± 2.65	9.22 ± 3.97	7.79 ± 4.10	<0.001 ^t^
Serum lactate (mmol/L)	2.15 [1.58; 3.72]	2.40 [1.4; 4.13]	2.20 [1.48; 4.12]	<0.926 ^u^
Deaths	7 (20.0)	30 (43.5)	37 (35.6)	0.02 ^c^ 3.08 (1.18–8.00)
Ventilator days (days)	3.00 [0.00; 11.0]	7.00 [4.00; 10.5]	6.00 [3.0; 10.75]	0.007 ^u^
ICU length of stay (days)	4.00 [2.50; 12.0]	7.00 [4.00; 11.0]	7.00 [4.00; 12.0]	0.054 ^u^
Hospital length of stay (days)	10.0 [7.00; 18.0]	14.0 [8.50; 22.0]	13.00 [8.25; 20.0]	0.11 ^u^

BMI: Body mass index; APACHE: Acute Physiology and Chronic Health Evaluation; SOFA: sequential organ failure assessment score; ICU: intensive care unit. Continuous data are presented as mean ± (standard deviation) or median [25th–75th percentile]; categorical data were presented as an absolute count (percentage). ^t^
*t*-test. ^u^ Mann–Whitney U test. ^c^ Pearson Chi-square test.

**Table 2 diagnostics-13-01427-t002:** Distributions of levels of nCD64, PCT, and WBC in the sepsis and non-sepsis groups.

Characteristics	Non-Sepsis Group *n* = 35	Sepsis Group *n* = 69	*p* *
nCD64 (molecules/cell)	745 [458; 906]	3106 [1970; 5200]	<0.001
PCT (ng/mL)	1.10 [0.30; 4.63]	43.4 [4.97; 70.30]	<0.001
WBC (G/L)	17.3 [13.1; 20.4]	13.9 [10.2; 18.5]	0.023

PCT: procalcitonin; WBC: white blood cell. Continuous data are presented as median [25th–75th percentile]. * Mann–Whitney U test.

**Table 3 diagnostics-13-01427-t003:** Sepsis diagnostic values of nCD64, PCT, and WBC among 104 study participants.

Biomarkers	AUC	95%CI	Cut-off	Sens	Spec	PPV	NPV
nCD64 (molecules/cell)	0.920	0.863–0.978	1311	89.9%	85.7%	92.5%	81.1%
PCT (ng/mL)	0.872	0.806–0.938	13.385	65.2%	93.9%	95.7%	56.4%
WBC (G/L)	0.637	0.526–0.748	17.19	73.9%	54.3%	76.1%	51.4%
nCD64 + PCT	0.924	0.868–0.981	-	88.4%	87.9%	93.8%	78.4%
nCD64 + WBC	0.906	0.845–0.967	-	78.3%	97%	98.2%	68.1%
nCD64 + WBC + PCT	0.919	0.863–0.975	-	79.7%	97%	98.2%	69.6%

PCT: procalcitonin; WBC: white blood cell; AUC: area under the curve; Sens: sensitivity; Spec: specificity; PPV: positive predictive value; NPV: negative predictive value.

## Data Availability

The datasets used and/or analyzed during the current study are available from the corresponding author on reasonable request.

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
