# Peer review of "Diagnostic Value of Neutrophil CD64 in Sepsis Patients in the Intensive Care Unit: A Cross-Sectional Study"

_diagnostics, 2023, doi:10.3390/diagnostics13081427_

Round 1
Reviewer 1 Report
The article deals with the dosage of nCD64 in cases of sepsis. In particular, here they want to validate the use of nCD64 in the diagnosis of sepsis. Indeed, in the database provided, nCD64 appears to have a role superimposable to that of PCT. Also, nCD64 appears to have higher sensitivity.
Please, in the introduction you could describe the new research methods for sepsis markers. For example, here you can add this to the bibliography: doi:10.3390/ijms23169354
The article is really well written. The experiments appear appropriate to demonstrate the effectiveness of the test. The statistical analysis is fine. I have only a question, have you tried any other quantitative/qualitative methods to detect sepsis status? Did you dosage nCD64 in case of septicemia or post-mortem?
Reviewer 2 Report
Dear authors,
I read your manuscript "DIAGNOSTIC VALUE OF NEUTROPHIL CD64 IN SEPSIS PATIENTS IN THE INTENSIVE CARE UNIT: A CROSS-SECTIONAL STUDY". I find the subject very interesting and useful.
I have the following comments:
Explain the acromims when they apear fiest inti the text, for exemple in abstract, line 14, 16 etc.
In Introduction section I believe everything is fine.
For the methods I suggest maybe to describe a little SOFA and APACHE scores.
In the result section, in line 157 in Table 1 I believe you must re-arrange table because p and OD together , lover and higher CI values may be difficult to follow.
Also result section may be improved.
The limitation of the study must be presented as a separate section.
Thank you
